# Heterogeneous Macrophage Activation in Acute Skeletal Muscle Sterile Injury and *mdx^5cv^* Model of Muscular Dystrophy

**DOI:** 10.3390/ijms26168098

**Published:** 2025-08-21

**Authors:** Xingyu Wang, Justin K. Moy, Yinhang Wang, Gregory R. Smith, Frederique Ruf-Zamojski, Pawel F. Przytycki, Stuart C. Sealfon, Lan Zhou

**Affiliations:** 1Department of Neurology, Hospital for Special Surgery, 535 East 70th Street, New York, NY 10021, USA; wangyin@hss.edu; 2Graduate Program in Bioinformatics, Boston University, 24 Cummington Mall, Boston, MA 02215, USA; jmmoy@bu.edu (J.K.M.); pawel@bu.edu (P.F.P.); 3Department of Neurology, Icahn School of Medicine at Mount Sinai, 1468 Madison Avenue, New York, NY 10029, USA; gregory.smith@mssm.edu (G.R.S.); frederique.ruf-zamojski@cshs.org (F.R.-Z.); stuart.sealfon@mssm.edu (S.C.S.); 4Faculty of Computing & Data Sciences, Boston University, 665 Commonwealth Avenue, Boston, MA 02215, USA

**Keywords:** inflammation, macrophages, muscle injury, muscular dystrophy

## Abstract

Monocytes/macrophages promote the repair of acutely injured muscle while contributing to dystrophic changes in chronically injured muscle in Duchenne muscular dystrophy (DMD) patients and animal models including *mdx* and *mdx^5cv^* mice. To elucidate the molecular mechanisms underlying this functional difference, we compared the transcriptomes of intramuscular monocytes/macrophages from *wild-typed* (*WT*) uninjured muscles, *WT* acutely injured muscles, and *mdx^5cv^* dystrophic muscles, using single cell-based RNA sequencing (scRNA-seq) analysis. Our study identified multiple transcriptomically diverse monocyte/macrophage subclusters, which appear to be induced by the intramuscular microenvironment. They expressed feature genes differentially involved in muscle inflammation, regeneration, and extracellular matrix (ECM) remodeling, but none of them conform to strict M1 or M2 activation. The Gpnmb^+^Spp1^+^ macrophage subcluster, an injury-associated subcluster that features the signature genes of reported scar-associated macrophages (SAMs) involved in ECM remodeling and fibrosis, is present transiently in acutely injured muscle and persistently in chronically injured dystrophic muscle, along with the persistence of monocytes. Our findings suggest that the persistent monocyte/macrophage infiltration and activation induced by continuous injury may underlie the pathogenic roles of macrophages in *mdx^5cv^* muscles. Controlling muscle injury and subsequent macrophage infiltration and activation may be important to the treatment of DMD.

## 1. Introduction

Skeletal muscle injury can be acute or chronic. Acute injury, caused by trauma or myotoxin exposure, usually repairs well with no or limited residual damage. Chronic injury, however, is usually associated with a chronic muscle disease, such as Duchenne muscular dystrophy (DMD), which cannot be completely repaired, as the muscle injury is caused by genetic defects in the dystrophin gene that leads to sarcolemma leakage, myofiber necrosis, chronic inflammation, and fibro-fatty tissue replacement [1]. Skeletal muscle injury repair is complex, involving inflammation, regeneration, and extracellular matrix (ECM) remodeling [2,3,4]. The inflammatory response, triggered by muscle injury, is predominated by monocyte/macrophage infiltration, which is one of the main determinants of injury repair outcomes [2,5,6,7].

Macrophages are heterogeneous and multi-functional cells, which can arise from both embryonic and postnatal hematopoiesis. Besides their originally identified roles in the activation and resolution of tissue inflammation, macrophages actively participate in multiple biological processes involved in tissue injury and repair [8,9,10,11,12]. Macrophages have been identified in both steady-state and injured skeletal muscles [13]. During acute skeletal muscle injury, blood Ly6C^hi^ inflammatory monocytes are quickly recruited, via CC chemokine receptor 2 (CCR2) signaling, by injured muscle where they differentiate into infiltrating macrophages [14,15]. Both resident and infiltrating macrophages are essential for the complete repair of acutely injured muscle, as depleting resident macrophages or blocking inflammatory monocyte recruitment compromises muscle regeneration [15,16,17]. In contrast, infiltrating macrophages are largely pathogenic which promote muscle fibrosis in chronically injured dystrophic muscles of *mdx* and *mdx^5cv^* mice [18,19], the mouse models of DMD, which is an X-linked lethal disease [20]. The molecular basis underlying the distinct roles played by macrophages in acutely and chronically injured muscles is not well understood. Elucidating such mechanisms would help guide the therapeutic manipulation of macrophages to promote skeletal muscle injury repair.

In the present study, we performed single cell-based RNA sequencing (scRNA-seq) to compare the transcriptomes of macrophages derived from uninjured and acutely injured muscles of *wild-typed* (*WT*) mice and those from chronically injured muscles of *mdx^5cv^* mice at 14 weeks of age. Our data showed that monocytes/macrophages expanded dramatically in response to skeletal muscle injury, and they were the predominant inflammatory cells in injured muscles. Multiple transcriptomically diverse monocyte/macrophage subclusters co-existed in each injured muscle sample. The monocyte, pro-inflammatory macrophage, and Gpnmb^+^Spp1^+^ macrophage subclusters appeared injury-associated, as they were barely detectable in uninjured muscle, but their fractions increased dramatically in injured muscles. The TLF^+^ (Timd4^+^ and/or Lyve1^+^ and/or Folr2^+^), MHCII^hi^, and interferon (IFN)-responsive macrophage (IFNRM) subclusters were present in both injured and uninjured muscles. Both Ly6C^hi^ and Ly6C^lo^ monocytes/macrophages were heterogeneous, containing different subclusters. None of these subclusters, however, conformed to strict M1 or M2 activation, based on their gene expression profiles. They differentially expressed genes involved in inflammation, regeneration, and extracellular matrix (ECM) remodeling. The Gpnmb^+^Spp1^+^ subcluster, featuring signature genes of scar-associated macrophages (SAMs) involved in ECM remodeling and fibrosis, was present in both acutely and chronically injured muscles, but only transiently in the former. Monocytes were also present transiently in acutely injured muscles but persistently in *mdx^5cv^* muscles. Our findings, in line with other published ones, suggest that persistent monocyte/macrophage infiltration and activation induced by chronic muscle injury appears to be the main factor that underlies the pathogenic function of macrophages in dystrophic muscles of *mdx^5cv^*.

## 2. Results

### 2.1. Intramuscular Monocytes/Macrophages Expand Dramatically in Injured Skeletal Muscle

Monocyte/macrophage infiltration predominates inflammatory response to both acute and chronic skeletal muscle injuries. In acutely injured muscle, Ly6C^hi^ inflammatory monocytes are recruited via CCR2 signaling, peaking at day 1 post injury (dpi) [14]. Within injured muscle, they differentiate into Ly6C^hi^ macrophages, phagocytose damaged tissues, and then switch into Ly6C^lo^ macrophages [14]. The number of Ly6C^lo^ macrophages and the number of total macrophages peak at 3–4 dpi [14,21]. Accompanying the Ly6C^hi^-to-Ly6C^lo^ switch, infiltrating macrophages undergo a drastic phenotype switch from pro-inflammatory to anti-inflammatory/pro-regenerative to support injury repair [11,14,21,22,23,24,25,26,27]. In chronically injured muscles associated with dystrophin deficiency in *mdx^5cv^* mice, muscle necrosis and inflammation start around 3 weeks of age, with both inflammation and fibrosis present at 14 weeks of age and persistent inflammation and progressive fibrosis mainly in the diaphragm afterwards [19,28]. To explore the molecular mechanisms underlying the functional differences in macrophages between acutely and chronically injured muscles, we performed scRNA-seq using single cell suspensions prepared from *WT* quadriceps 1 day and 3 days after barium chloride (BaCl_2_)-induced acute injury, quadriceps and diaphragm muscles of *mdx^5cv^* mice, and uninjured control quadriceps and diaphragm muscles of *WT* mice, all at 14 weeks of age. The single cell suspensions contain mononuclear cells but not myofibers, as polynuclear myofibers cannot be included due to their large size. Each sample was pooled from five male mice to minimize individual variations. All six samples were analyzed simultaneously (see Appendix A for quality control).

After filtering out the cells of low quality and the genes with zero read count [29], we obtained reads of 16,766 genes from 3814 cells of *WT* quadriceps, 17,677 genes from 6988 cells of *WT* diaphragm, 18,398 genes from 6302 cells of *mdx^5cv^* quadriceps, 18,152 genes from 6089 cells of *mdx^5cv^* diaphragm, 19,231 genes from 6813 cells of day 1 quadriceps, and 20,942 genes from 4875 cells of day 3 quadriceps for analysis. The automated annotation of main cell types was performed using SingleR with the ImmGen reference set [30,31]. Multiple clusters resembling different cell types, including monocytes/macrophages, were identified from each sample (Figure 1A). The proportion of monocytes/macrophages was very low in uninjured *WT* quadriceps and diaphragm but was markedly increased in all injured muscle samples (Figure 1B). They became the largest cell population in acutely injured quadriceps and in *mdx^5cv^* quadriceps (Figure 1B). We next determined the density of intramuscular monocytes/macrophages by flow cytometry (FACS) analysis. Intramuscular monocytes/macrophages were identified as CD45^+^CD64^+^F4/80^+^ cells (Figure 1C). The density was calculated based on the percentage of monocytes/macrophages to the total intramuscular single cells, total number of intramuscular single cells, and mass of whole muscle. Monocyte/macrophage density was markedly increased in all injured muscles compared to uninjured controls (Figure 1D), with the highest density observed in *WT* quadriceps at 3 dpi (Figure 1D), consistent with previous reports [14,21].

### 2.2. Transcriptomically Diverse Subclusters of Monocytes/Macrophages Co-Exist in Normal and Injured Skeletal Muscles

We next addressed the differences in the transcriptome diversity of monocytes/macrophages in uninjured, acutely injured, and chronically injured muscles. To this end, the transcriptome data of the monocyte/macrophage cell populations was selected for comparisons. Monocytes/macrophages were very few in uninjured muscles and their numbers were too low for reliable analysis (*WT* quadriceps: 83 cells; *WT* diaphragm: 61 cells). We therefore replaced the *WT* data generated by this whole-muscle scRNA-seq study with the *WT* data derived from our previously published monocyte/macrophage scRNA-seq study identifying resident macrophages in uninjured skeletal muscles (GEO: GSE142480) [13]. Inclusion of this dataset also provides a reference for analyzing the resident macrophage population in injured skeletal muscles. The two sets of scRNA-seq data were first integrated to correct batch differences, and the subclusters of monocytes/macrophages with different transcriptomes were then analyzed.

Our analysis generated seven subclusters (Figure 2A,B). The top 50 differentially expressed genes (DEGs) of each subcluster were listed in Appendix A, and a complete list of all DEGs of each subcluster was provided in Appendix A. Among the seven subclusters, three were primarily present in the injured muscle samples, as they were minimally present in the two uninjured muscle samples, including a subcluster resembling Ly6C^hi^ monocytes featuring high expressions of *Plac8*, *Ly6c2*, and *Ccr2*, a pro-inflammatory macrophage subcluster featuring high expressions of pro-inflammatory genes, and a Gpnmb^+^Spp1^+^ macrophage subcluster featuring the signature genes of reported scar-associated macrophages (SAMs) [32,33,34] (Figure 2A–D, Appendix A). DEGs of the pro-inflammatory subcluster were highly enriched in genes involved in inflammation, as 14 out of top 20 DEGs were inflammatory genes, including *Cxcl3*, *Il1b*, *Arg1*, *Cxcl1*, *Thbs1*, *Ppbp*, *Clec4e*, *Ptgs2*, *Cd14*, *Ier3*, *Ccl2*, *Ccl6*, *Ccl7*, *Chil3*, and *Il1a* (Appendix A). Functional enrichment analysis further showed that the DEGs of the pro-inflammatory subcluster were enriched in pathways involved in acute inflammatory and stress responses (Appendix A). DEGs of the Gpnmb^+^Spp1^+^ subcluster featured the signature genes of SAMs which were originally identified as CD9^+^TREM2^+^ macrophages and found to have a similar phenotype in multiple tissues and organs with physical proximity to excessive ECM [32,33,34]. It has been proposed that differential expressions of at least five of six markers, including *Gpnmb*, *Spp1*, *Fabp5*, *Trem2*, *Cd9*, and *Cd63*, are required to define SAMs [32]. All these six markers were among the top 15 DEGs of the Gpnmb^+^Spp1^+^ subcluster identified by our current study (Figure 2C and Appendix A), indicating that this subcluster resembles SAMs. Functional enrichment analysis showed that DEGs of this subcluster were enriched in pathways of lipid-related cellular activities, PPAR signaling, phagocytosis, energy metabolism, and antigen processing-cross presentation (Appendix A). Similar macrophage subtypes were also identified by others in both acutely injured muscle and dystrophic muscle. Coulis et al. reported in 4–6-week-old *mdx* mice a Lgals3^+^Spp1^+^ macrophage subtype associated with stromal cells and ECM [35], of which 5 out of top 10 DEGs (*Spp1*, *Trem2*, *Fabp5*, *Trem2*, and *Syngr1*) were also among the top 10 DEGs of the Gpnmb^+^Spp1^+^ subcluster identified by the current study (Figure 2C and Appendix A). The authors suggested that this macrophage subtype is pro-fibrotic, expressing *Spp1* and interacting with stromal cells [35]. Patsalos et al. reported a growth factor-expressing macrophages (GFEMs) in day 4 cardiotoxin-injured muscle and 2-month-old dystrophic muscle of *D2-mdx* mice [36,37]. This GFEM subset featured expressions of *Gpnmb*, *Gdf15*, *Syngr1*, *Fabp4*, *Cd36*, *Slc6a8*, and *Uap111*, which were also among the top 30 DEGs of the Gpnmb^+^Spp1^+^ subcluster identified from the current study (Appendix A). Spatial transcriptome analysis further showed that this Gpnmb^+^ GFEM subset was localized between inflammatory lesions and healthy/regenerating areas of dystrophic muscle and might promote regeneration by producing pro-regenerative growth factors, GDF-15 and IGF-1 [37]. Taken together, the Gpnmb1^+^Spp1^+^ macrophage subcluster identified here resembles the previously identified macrophage subtypes that appear involved in regulating ECM remodeling and muscle regeneration.

The fraction of these injury-associated monocyte/macrophage subclusters varied among different muscle samples (Figure 2B,D, and Appendix A). Monocytes were readily identified in all four injured muscle samples. In quadriceps post-acute injury, the fraction of monocytes on day 3 was greater than that on day 1 (Figure 2D and Appendix A), indicating the continuous infiltration of blood monocytes during the first 3 days post-injury. In *mdx^5cv^* muscles, the monocytes fraction was greater in the diaphragm than in the quadriceps (Figure 2D and Appendix A). The pro-inflammatory and Gpnmb^+^Spp1^+^ subclusters were barely identifiable in uninjured *WT* quadriceps or diaphragm, but they were prominent in injured muscles (Figure 2B,D, and Appendix A). The pro-inflammatory subcluster was the largest subcluster on day 1 but was significantly reduced on day 3, despite the continuous expansion of monocytes (Figure 2D and Appendix A). This cluster was much smaller in *mdx^5cv^* muscles (Figure 2D). The findings suggest a unique strong pro-inflammatory microenvironment in day 1 quadriceps, presumably caused by massive muscle necrosis, to drive the differentiation of infiltrating monocytes into pro-inflammatory macrophages. The Gpnmb^+^Spp1^+^ subcluster was prominent in day 3 quadriceps and in *mdx^5cv^* quadriceps and diaphragm (Figure 2D and Appendix A).

Different from the above three injury-associated monocyte/macrophage subclusters, the other four macrophage subclusters were present in both uninjured and injured muscle samples, including an IFNRM subcluster, a proliferating subcluster, and two resident-like subclusters. The IFNRM subcluster featured high expressions of IFN-responsive genes (ISGs) such as *Rsad2*, *Ifit1*, *Cxcl10*, and *Irf7* (Figure 2C and Appendix A). Sixteen out of the top 20 DEGs of this subcluster were known ISGs (Appendix A), indicating an active response to IFN signals. A highly similar IFNRM subtype, featuring high expressions of *Ifit3*, *Ifi204*, *Isg15*, *Ifitm3*, *Irf7*, *Plac8*, and *Rsad2*, was previously identified and reported to promote the proliferation and differentiation of satellite cells via the production of CXCL10 in acutely injured muscle [38]. A similar IFNRM subtype was also identified in the dystrophic muscle of 4-week-old *mdx* mice [35]. Functional enrichment analysis further showed that DEGs of this subcluster were highly enriched in type 1/2 IFN signaling, along with the pathways of endocytosis, endogenous damage-associated molecular pattern (DAMP) sensing (NOD-like receptor signaling, RIG-I/MDA-5 signaling, and cytosolic DNA sensing), and IL-10 signaling (Appendix A). The proliferating subcluster featured high expressions of cell-cycle genes such as *Stmn1*, *Mki67*, *Hmgb2*, and *Cdk1* (Figure 2C and Appendix A).

Recently, three subclusters of resident macrophages were identified, co-existing across different tissues and defined by their distinct expression of *Timd4*, *Lyve1*, *Folr2*, *Ccr2*, and MHCII genes [39,40]. They are Timd4^+^ and/or Lyve1^+^ and/or Folr2^+^ (TLF^+^) Ccr2^−^MHCII^lo^ subcluster, TLF^−^Ccr2^+^MHCII^hi^ subcluster, and TLF^−^Ccr2^−^MHCII^hi^ subcluster [39,40]. The TLF expression primarily marks the embryo-derived resident macrophages [39,40], which can persist into adulthood through proliferative self-renewal. Similarly, we previously identified in steady-state skeletal muscle the existence of two subsets of resident macrophages: MHCII^hi^Lyve1^lo^ and MHCII^lo^Lyve1^hi^ macrophages [13]. The MHCII^hi^Lyve1^lo^ subset almost exclusively originates from hematopoietic stem cells (HSCs), while the MHCII^lo^Lyve1^hi^ subset is contributed significantly by embryonic hematopoietic progenitors of non-HSC origin [13]. With the inclusion of our previously published data analyzing resident macrophages in uninjured muscle (GEO: GSE142480) [13], two resident-like macrophage subclusters were identified by the current study: 1) the MHCII^hi^ subcluster featuring high expressions of *Cx3cr1* and MHCII genes such as *H2-Aa*; 2) the TLF^+^ subcluster featuring high expressions of *Timd4*, *Lyve1*, *Folr2*, and *Cd163* (Figure 2C and Appendix A). Further, *Ccr2* was barely expressed by the TLF^+^ subcluster while it was highly expressed by the MHCII^hi^ cluster (Figure 2E). *Ccr2* was highly expressed by the monocytes, IFNRM, and pro-inflammatory subclusters, and was moderately expressed by the Gpnmb^+^Spp1^+^ and proliferating subclusters (Figure 2E). Conversely, *Timd4* and *Lyve1* were primarily expressed by the TLF^+^ cluster (Figure 2E). Since CCR2 is expressed by Ly6C^hi^ blood monocytes [14,41] and this expression is required for the Ly6C^hi^ blood monocytes to be recruited by skeletal muscle [15], our findings suggest that the non-TLF^+^ macrophage subclusters may contain a monocytic origin. This hypothesis awaits further validation by lineage tracing.

The IFNRM subcluster was not identifiable in day 1 quadriceps (Figure 2B,D, and Appendix A). The TLF^+^ subcluster was the predominant subcluster in both the *WT* quadriceps and diaphragm, with its fraction decreased dramatically in all injured muscles (Figure 2D and Appendix A), likely due to being outnumbered by the other subclusters derived from infiltrating monocytes/macrophages. In both *mdx^5cv^* quadriceps and diaphragm, the MHCII^hi^ subcluster was the predominant subcluster (Figure 2D and Appendix A), with its fraction increased dramatically in *mdx^5cv^* quadriceps compared to *WT* quadriceps (Figure 2D and Appendix A). The proliferating subcluster showed the greatest fraction in day 3 quadriceps (Figure 2D and Appendix A), suggesting the more active proliferation of intramuscular macrophages in acutely injured muscles at this stage than others.

The difference in the proportion of individual monocyte/macrophage subclusters in different muscle samples may reflect the difference in the tissue microenvironment that drives the activation and differentiation of macrophages. We next inferred the potential differentiation trajectory among the identified monocyte/macrophage subclusters by pseudotime analysis with both Monocle 3 and Slingshot (Figure 3A,B). Monocyte was set as the starting point. The TLF^+^ subcluster was the most distal to the monocytes subcluster, which was consistent with the presumed non-monocytic origin of this subcluster. Interestingly, compared with the Gpnmb^+^Spp1^+^ subcluster which was present on day 3 but not day 1 post-acute injury, the pro-inflammatory subcluster, which predominated monocytes/macrophages on day 1, was more distal to the monocytes. The finding suggests that the Gpnmb^+^Spp1^+^ subcluster may be differentiated directly from the monocytes rather than through the pro-inflammatory subcluster. Compared with the Gpnmb^+^Spp1^+^ subcluster, the IFNRM and MHCII^hi^ subclusters were both more distal to the monocytes (Figure 3A,B). 

In summary, multiple transcriptomically diverse monocyte/macrophage subclusters co-exist in uninjured, acutely injured, and chronically injured skeletal muscles, among which the monocytes, pro-inflammatory, and Gpnmb^+^Spp1^+^ subcluster are injury-associated subclusters. The macrophage composition is very different in day 1 quadriceps compared to day 3 quadriceps and *mdx^5cv^* quadriceps and diaphragm, with a unique presence of a large pro-inflammatory subcluster in the former. The Gpnmb^+^Spp1^+^ subcluster was similarly identified in day 3 quadriceps and *mdx^5cv^* quadriceps and diaphragm.

### 2.3. Both Ly6Chi and Ly6Clo Intramuscular Macrophages Are Heterogeneous

During acute skeletal muscle injury, infiltrating monocytes/macrophages undergo a Ly6C^hi^-to-Ly6C^lo^ phenotype switch [14,21], which can be detected by FACS analysis of cell-surface expression of Ly6C. As shown in Figure 4A, most monocytes/macrophages were Ly6C^hi^ on day 1 while were Ly6C^lo^ on day 3 post-injury. In *mdx^5cv^* quadriceps and diaphragm, both Ly6C^hi^ and Ly6C^lo^ macrophages were present, with the Ly6C^lo^ subpopulation being the predominant one (Figure 4A). Since our scRNA-seq analysis identified multiple monocyte/macrophage subclusters in each of these four samples (Figure 2), we next addressed which subclusters were Ly6C^hi^ or Ly6C^lo^ by analyzing the expression of the *Ly6c2* gene that encodes Ly6C (Figure 4B). Only monocytes and IFNRM subclusters showed a high expression level of *Ly6c2* across all four samples (Figure 4B). Interestingly, the pro-inflammatory subcluster, which was the predominant subcluster in day 1 quadriceps (Figure 3B), showed a low expression of *Ly6c2* (Figure 4B) despite the majority of monocytes/macrophages at this time point being Ly6C^hi^ as shown by FACS (Figure 4A). The finding suggests that the pro-inflammatory subcluster has already downregulated the *Ly6c2* gene expression in day 1 quadriceps. The TLF^+^ subcluster was also identified in day 1 quadriceps (Figure 2B,D), which barely expressed *Ly6c2*, belonging to the Ly6C^lo^ subpopulation (Figure 4B). In day 3 quadriceps, the pro-inflammatory subcluster was diminished (Figure 2B,D). The Ly6C^hi^ monocytes/macrophages were predominantly monocytes with a minor contribution from the IFNRM subcluster. The Ly6C^lo^ macrophages mainly consisted of the Gpnmb^+^Spp1^+^ and proliferating subclusters (Figure 2D and Figure 4B). In both the quadriceps and diaphragm of 14-week-old *mdx^5cv^* mice, the Ly6C^hi^ subpopulation contained monocytes and IFNRM subclusters, while the Ly6C^lo^ subpopulation contained mainly Gpnmb^+^Spp1^+^ and MHCII^hi^ subclusters with small percentages of TLF^+^ and proliferating subclusters (Figure 2D and Figure 4B). In summary, both Ly6C^hi^ and Ly6C^lo^ intramuscular monocytes/macrophages are heterogeneous, containing transcriptomically different subclusters with varying fractions in different injured muscle samples.

### 2.4. Monocyte/Macrophage Subclusters Do Not Conform to Strict M1 or M2 Activation

Based on the activation status, macrophages were historically classified into M1 (classically activated) and M2 (alternatively activated) subtypes, which was originally defined by in vitro studies and in vivo studies of parasite infections [42]. M1 and M2 macrophages differ in their activation stimuli, cell surface markers, cytokine production profiles, arginine metabolism, and activated transcriptional factors [42,43,44]. M1 activation is pro-inflammatory, while M2 can be anti-inflammatory, pro-regenerative, and/or pro-fibrotic. However, our previously published study showed that both Ly6C^hi^ and Ly6C^lo^ macrophages expressed mixed M1/M2 genes on day 1 and day 3 after acute muscle injury [21], challenging this bi-polar activation paradigm. Since scRNA-seq showed heterogeneity of both Ly6C^hi^ and Ly6C^lo^ macrophages in injured skeletal muscles (Figure 4), we would address whether the mixed M1/M2 gene expression could be due to the co-existence of macrophage subclusters conforming to different M1/M2 activation status. To elucidate this, we examined the expression of reported M1 and M2 genes [42,43,44] by different monocyte/macrophage subclusters (Figure 5).

The Ly6C^hi^ monocytes/macrophages in day 1 quadriceps mainly contained monocytes and pro-inflammatory macrophage subclusters (Figure 3). Both subclusters highly expressed M1 markers *Tnf*, *Il1b*, *Nfkbiz*, and *Irf5* (Figure 5A) and simultaneously expressed M2 markers *Mrc1*, *Il4ra*, *Tgfb1*, *Arg1*, *Socs3*, and *Stat6* (Figure 5B). Neither was strictly M1. Interestingly, the TLF^+^ macrophage subcluster, which was the predominant Ly6C^lo^ cluster on day 1 post-acute injury (Figure 3 and Figure 4), also expressed a high level of both M1 and M2 genes, similarly to the Ly6C^hi^ subclusters (Figure 5). The findings suggest that the microenvironment in day 1 injured muscle drives mixed M1/M2 gene expression by all the intramuscular monocyte/macrophage subclusters. The mixed M1/M2 gene expression was also detected in monocyte/macrophage subclusters on day 3 post-acute injury. However, both monocytes and pro-inflammatory subclusters downregulated some M1 genes, including *Tnf*, *Il1b*, *Nfkbiz*, and *Irf5*, and M2 genes, including *Arg1* and *Socs3*, on day 3 as compared to day 1 (Figure 5). The expression of the other M2 genes, such as *Mrc1*, *Il4ra*, *Tgfb1*, and *Stat6*, was either unchanged or slightly increased (Figure 5B). These changes are likely to reflect the changes in the tissue microenvironment. The IFNRM and Gpnmb^+^Spp1^+^ subclusters were sufficiently identified on day 3 but not day 1 (Figure 3). Both also expressed mixed M1/M2 genes (Figure 5) and thus did not conform to strict M1 or M2 activation. Overall, a general shift from M1-biased to M2-biased activation was observed in intramuscular monocytes/macrophages from day 1 to day 3, as evidenced by a decrease in the expression of *Tnf*, *Il1b*, and *Nfkbiz* and an increase in the expression of *Mrc1*, *Tgfb1*, and *Igf1* (Figure 5). The activation status of monocytes/macrophages in the two *mdx^5cv^* samples was largely similar to those in quadriceps on day 3 post-injury, with a slightly further bias towards M2 activation status (Figure 5). Notably, the monocyte/macrophage subclusters identified in *WT* uninjured muscles all expressed a very high level of M2 marker genes including *Retnla*, *Cd163*, *Tgfb1*, and *Igf1* (Figure 5), suggesting a pro-M2 microenvironment in normal steady-state skeletal muscle.

In summary, our data indicates that intramuscular monocyte/macrophage subclusters do not conform to strict M1 or M2 activation. Rather, their activation status varies depending on the changes in the microenvironment of injured skeletal muscle. The classical bi-polar activation paradigm does not apply to the complex in vivo monocyte/macrophage activation status in either acutely or chronically injured muscle.

### 2.5. Different Monocyte/Macrophage Subclusters Express Genes Differentially Involved in Muscle Inflammation, Regeneration, and ECM Remodeling

Macrophages play a multifaceted role in skeletal muscle injury and repair, regulating multiple processes including inflammation, regeneration, and ECM remodeling [27]. We next analyzed by dot plots the expression of the genes involved in inflammation, regeneration, and ECM remodeling (Figure 5 and Figure 6) to address the potential contribution of individual subclusters to these processes.

The expression of pro-inflammatory genes, including *Tnf*, *Il1b* (Figure 5A), *Il1a*, *Ccl2*, and *Cxcl2* (Figure 6), was much higher by the monocytes and pro-inflammatory macrophage subclusters than by the others, especially in day 1 quadriceps. These two subclusters also expressed an exclusively high level of *Arg1*, an inflammatory regulatory gene [45], in day 1 quadriceps (Figure 5B). The monocytes and pro-inflammatory clusters are thus highly inflammatory.

Skeletal muscle regeneration following injury relies on myogenic stem cells, muscle satellite cells (MuSCs) [46,47]. Injury-induced changes in the muscle microenvironment activate MuSCs to regenerate muscle fibers, a process consisting of the activation, proliferation, and differentiation of MuSCs [48,49,50]. Macrophages at early stage post-injury have been reported to produce a high level of soluble factors known to stimulate the activation and proliferation of MuSCs, including IL-6 [51], TNF-α [52], and fibronectin [53]. Although *Il6* was barely expressed by any monocyte/macrophage subcluster (Figure 5A), both *Tnf* (encoding TNF-α, Figure 5A) and *Fn1* (encoding fibronectin, Figure 6) were predominantly expressed by monocytes and pro-inflammatory subclusters, especially on day 1, suggesting a role for these two subclusters in promoting MuSC activation and proliferation besides inflammation. Macrophage-derived IGF-1 was shown to stimulate MuSC differentiation and myofiber growth [15,54,55]. The *Igf1* gene was barely expressed by the monocytes or pro-inflammatory subcluster (Figure 5B) but was expressed at a high level by the Gpnmb^+^Spp1^+^ subcluster in day 3 quadriceps and in *mdx^5cv^* quadriceps and diaphragm (Figure 5B), consistent with the findings by Patsalos et al. [37]. Additionally, the MHCII^hi^ and TLF^+^ subclusters also expressed a considerable level of *Igf1* in the same samples (Figure 5B). Given the high percentage of the Gpnmb^+^Spp1^+^ subcluster in day 3 quadriceps and the high percentages of the Gpnmb^+^Spp1^+^, MHCII^hi^, and TLF^+^ subclusters in *mdx^5cv^* quadriceps and diaphragm (Figure 2D), these subclusters appear to be the primary macrophage sources of IGF-1 which can promote muscle regeneration [15,54,55].

Recently, several macrophage-derived molecules were reported to promote skeletal muscle regeneration following injury, including CXCL10 [38], NAMPT [56], GDF3 [57,58], and GDF15 [36]. Significantly, *Cxcl10* was almost exclusively expressed by the IFNRM subcluster and *Nampt* was expressed at the highest level by the IFNRM subcluster (Figure 6). This subcluster also expressed the highest level of *Il10* (Figure 5B), an anti-inflammatory cytokine shown to promote muscle injury repair [59,60]. The findings suggest that this subcluster, although small, might play a role in promoting muscle regeneration following injury. On the other hand, *Gdf15* was expressed at the highest level by the Gpnmb^+^Spp1^+^ subcluster in day 3 quadriceps and *mdx^5cv^* muscles, and by the TLF^+^ subcluster in day 1 quadriceps (Figure 6) where the Gpnmb^+^Spp1^+^ subcluster was largely absent (Figure 2D). The *Gdf3* expression was generally low in all injured muscles, with the highest expression seen in day 3 quadriceps by the Gpnmb^+^Spp1^+^ subcluster (Figure 6). The findings might imply a role for the Gpnmb^+^Spp1^+^ subcluster in muscle regeneration.

Fibro/adipogenic progenitors (FAPs) are the primary intramuscular fibrogenic cells that produce ECM [61,62,63,64]. Macrophages play important roles in regulating FAPs. Macrophage-derived TNF-α can induce apoptosis of FAPs to limit excessive FAP accumulation and subsequent ECM production [65]. On the other hand, macrophages can also produce potent pro-fibrotic factors, including osteopontin (encoded by *Spp1*) and TGF-β1, to stimulate the proliferation and differentiation of fibrogenic cells [66,67]. The monocytes and pro-inflammatory subclusters showed the highest expression of *Tnf* (Figure 5A), while the Gpnmb^+^Spp1^+^ subcluster showed the highest expression of *Spp1* (Figure 6). Interestingly, *Tgfb1* was ubiquitously expressed by all the monocyte/macrophage subclusters (Figure 5B). The Gpnmb^+^Spp1^+^ subcluster also expressed the highest level of *Timp2*, a gene that encodes tissue inhibitors of metalloproteinase-2 (TIMP-2) (Figure 6). TIMP-2 inhibits the activity of matrix metalloproteinase-14 (MMP-14) [68] that is required for the activation of latent TGF-β1 [69]. TIMP-2 is also required for the activation of MMP-2 to degrade ECM components [70]. The simultaneous high expression of both *Spp1* and *Timp2* by the Gpnmb^+^Spp1^+^ subcluster suggests that these macrophages might play a significant role in tuning ECM remodeling in injured muscle, a process that is required to provide a structural support for the repair of acutely injured muscle [71] but also contributes to progressive fibrosis in chronically injured muscle [72]. By expressing a considerable level of *Igf1*, the MHCII^hi^ and TLF^+^ subclusters might also be involved in ECM remodeling, as IGF-1 can display both pro-regenerative [15,54,55] and pro-fibrotic functions [73,74]. Taken together, our scRNA-seq analysis suggests that the transcriptomically different monocyte/macrophage subclusters may exert different functions in injured muscles, with respect to muscle inflammation, regeneration, and ECM remodeling.

### 2.6. Reparative Regeneration of Acutely Injured Muscle Is Accompanied by Disappearance of Gpnmb^+^Spp1^+^ Subcluster and Emergence of Inflammation Resolution Subclusters at Late Stages of Injury Repair

Gpnmb^+^Spp1^+^ macrophages have been shown to be important for ECM remodeling and fibrosis [32,33,35,75,76,77,78,79], and they are present in both day 3 acutely injured quadriceps and *mdx^5cv^* muscles (Figure 2). Since the increased ECM deposition is transient in acutely injured muscle while persistent in *mdx^5cv^* muscles, we addressed whether the Gpnmb^+^Spp1^+^ macrophage subcluster recedes at late stages (after 3 dpi) of acutely injured muscle, corresponding to the regenerative repair. We used previously published scRNA-seq data generated from a study of acutely injured tibialis anterior (TA) muscle, in which whole TA muscle cells were collected from multiple time points (0.5, 2, 3.5, 5, 10, and 21 days) post injury, covering the entire process of acute muscle injury repair [80]. The data of the monocyte/macrophage population from this study was first sorted by clustering, integrated to correct batch differences, and then subjected to analysis for transcriptomically different clusters of monocytes/macrophages.

As shown in Figure 7A, a total of nine subclusters were identified (Figure 7A, Appendix A), among which the monocytes, pro-inflammatory, Gpnmb^+^Spp1^+^, and IFNRM subclusters were similar to the corresponding subclusters identified in acutely injured quadriceps muscle at early stages (Figure 2), based on the comparison of their DEGs (Appendix A). Two MHCII^hi^ subclusters were identified in this new analysis, with both featuring DEGs of MHCII molecules (*H2-Aa*, *H2-Ab1*, *H2-Eb1*, and *Cd74*) (Figure 7A and Appendix A). One MHCII^hi^ cluster was present in both uninjured and injured muscles and was thus named MHCII^hi^-resident subcluster. The other MHCII^hi^ subcluster was present only in injured muscles on and after 3.5 days post-injury, likely differentiated from infiltrating macrophages following injury (Figure 7A,B). Three new subclusters were identified in this analysis: Spp1^+^Arg1^+^, Gpx3^+^, and Ccl8^+^ subclusters. The Spp1^+^Arg1^+^ subcluster shared DEGs with both the Gpnmb^+^Spp1^+^ subcluster (*Spp1*, *Fabp5*, and *Cd9*) and monocytes and pro-inflammatory subclusters (*Arg1*, *Fn1*, *S100a4*, *Pf4*, and *Ccl2*) (Figure 7A, Appendix A). This subcluster was almost exclusively identified on day 2 post-injury (Figure 7A,B), suggesting that it may represent an intermediate state between monocyte/pro-inflammatory macrophage subclusters and the Gpnmb^+^Spp1^+^ subcluster. Both Gpx3^+^ and Ccl8^+^ subclusters emerged only on and after day 3.5 post-injury (Figure 7A,B). The Gpx3^+^ subcluster featured DEGs promoting inflammation resolution (*Gpx3*, *Rgs2*, and *Pink1*) [81,82,83] (Appendix A). The Ccl8^+^ subcluster featured *Ccl8* and M2-biased genes (*Cbr2*, *Folr2*, and *Mrc1*). *Ccl8* has been reported to be highly expressed by tumor-associated macrophages promoting an immunosuppressive status [84]. Therefore, both Gpx3^+^ and Ccl8^+^ subclusters might potentially play a role in promoting the resolution of inflammatory response to allow the completion of reparative regeneration. 

The fraction of each monocyte/macrophage subcluster in acutely injured skeletal muscle changed along with the repair process (Figure 7B). As we speculated, the Gpnmb^+^Spp1^+^ subcluster started to recede on day 5 (Figure 7B). Meanwhile, the Gpx3^+^ and Ccl8^+^ subclusters emerged on day 3.5 post-injury and persisted into day 21 post-injury (Figure 7B). The temporal changes in these subclusters correspond to the resolution of transient fibrosis and inflammation. The MHCII^hi^ macrophages (including MHCII^hi^ and MHCII^hi^-resident subclusters) gradually became the predominant subtype at later stages, day 10 and day 21 post-injury (Figure 7B). In contrast, the Gpnmb^+^Spp1^+^ subcluster co-existed with the MHCII^hi^ subcluster in both the quadriceps and diaphragm of 14-week-old *mdx^5cv^* mice (Figure 3). As mentioned before, a similar macrophage subtype was also identified in *mdx* muscle at 4–6 weeks of age [35] and in *D2-mdx* muscle at two months of age [37]. Therefore, Gpnmb^+^Spp1^+^ macrophages appear to persist in dystrophic muscles and possibly contribute to muscle fibrosis. Expectedly, the Gpx3^+^ and Ccl8^+^ subclusters were only identified in the late stages of acutely injured skeletal muscle but not in *mdx^5cv^* muscles where chronic inflammation was present.

## 3. Discussion

The transcriptome diversity and dynamics of monocytes/macrophages have been studied using scRNAseq analysis in acute skeletal muscle injury [36,80,85,86,87] and DMD mouse models [35,37,88,89,90,91]. However, the fundamental question why monocytes/macrophages promote the repair of acutely injured muscle while contributing to dystrophic changes in chronically injured muscle remained unanswered. By a direct comparison of the transcriptomes of intramuscular monocytes/macrophages among uninjured, acutely injured, and chronically injured dystrophic muscles, our present scRNA-seq analysis showed no pathogenic monocyte/macrophage subcluster exclusively identified in dystrophic muscle. Instead, our data, along with others’ [35,37], demonstrate that the injury-associated monocytes and Gpnmb^+^Spp1^+^ macrophage subclusters persist in dystrophic muscle but not in acutely injured muscle. The persistence of these two subclusters, which reflects continuous monocyte infiltration and differentiation that is triggered by constant muscle injury in dystrophic muscle, might contribute to chronic inflammation and progressive fibrosis.

The Gpnmb^+^Spp1^+^ macrophages feature the expression of signature genes of SAMs, including *Gpnmb*, *Spp1*, *Fabp5*, *Trem2*, *Cd9*, and *Cd63*, which are reportedly associated with fibrogenesis across different tissues [32,33,34,75,76,77,78,79]. Osteopontin, encoded by *Spp1*, is a highly phosphorylated glycoprotein that has been increasingly recognized for its involvement in the progression of tissue fibrosis [92]. Lack of osteopontin in *mdx* mice by the global genetic deletion of *Spp1* led to reduced intramuscular TGF-β, reduced fibrosis in the diaphragm, and increased muscle regeneration and strength [93], demonstrating an in vivo pro-fibrotic role for this protein. Moreover, the ablation of macrophage-specific *Spp1* in *mdx* led to a reduction in an adipogenic stromal cell population, reduced intramuscular fat accumulation, and improved muscle function [94]. These in vivo findings support the notion that Gpnmb^+^Spp1^+^ macrophages may contribute to the progressive fibrofatty tissue replacement in dystrophic muscle via, at least in part, the osteopontin expression. In line with this hypothesis, spatial transcriptome analysis has shown that Gpnmb^+^Spp1^+^ macrophages are associated with stromal cells and ECM genes in the dystrophic muscle of *mdx* mice [35]. Besides *Spp1*, the macrophage-specific deletion of *Trem2*, another gene differentially expressed by Gpnmb^+^Spp1^+^ macrophages, has been reported to reduce fibrosis in mouse models of pulmonary fibrosis [95] and myocardial infarction [96]. These reports added additional support to the potential fibrogenic roles of Gpnmb^+^Spp1^+^ macrophages in injured skeletal muscle. The Gpnmb^+^Spp1^+^ subcluster identified in the present study represents a significant portion of intramuscular macrophages in both dystrophic muscles and acutely injured muscles from day 3 to day 5, suggesting that this subcluster might contribute not only to the progressive fibrofatty tissue replacement in dystrophic muscle, which is pathogenic, but also to the transient fibrosis in acutely injured muscle, which is a normal process that provides a structural support to acute injury repair. The receding of this subcluster after day 5 might be important to the resolution of the transient fibrosis in acutely injured muscle for the completion of injury repair.

Gpnmb^+^Spp1^+^ macrophages appear multifunctional. They also expresses the highest level of pro-regenerative growth factors, *Igf1* [15,54,55] and *Gdf15* [36], both in day 3 quadriceps and *mdx^5cv^* muscles, suggesting a role in pro-regeneration, which is also suggested by previous reports. Patsalos et al. showed that Gpnmb⁺ macrophage subsets promote muscle regeneration by producing GDF15 [36]. Global Gpnmb knockout impairs muscle regeneration following CTX-induced acute injury [37]. The Gpnmb⁺ macrophage subset resides with regenerating myofibers in both day 4 acutely injured muscle and dystrophic muscle of 2-month-old *D2-mdx* mice as revealed by a spatial transcriptomics study [37]. These findings suggest that in addition to fibrogenesis, Gpnmb^+^Spp1^+^ macrophages may also play a role in myofiber regeneration. Macrophage-specific targeting of the core signature genes of this macrophage subtype is required to precisely define its roles in both acutely and chronically injured muscles. Consistent with the findings from the DMD mouse models, the macrophage subtype that expresses *Spp1* and *Gpnmb* was increased in the dystrophic muscle of DMD patients as compared with healthy controls [97].

Besides the Gpnmb^+^Spp1^+^ subcluster, the present study also identified multiple other transcriptomically diverse monocyte/macrophage subclusters in both acutely and chronically injured skeletal muscles. The monocytes and pro-inflammatory macrophage subclusters appear to be pro-inflammatory, given their very high expression of *Tnf* and *Il1b.* They might also contribute to muscle regeneration because they express a high level of *Tnf* and *Fn1*, which can stimulate the activation and proliferation of MuSCs [52,53]. The IFNRM subcluster features a high-level expression of IFN-responsive genes, along with the genes involved in endogenous DAMP sensing. DEGs of this subcluster are also enriched in pathways of endocytosis and antigen processing. The findings suggest that IFNRMs might participate in phagocytosing apoptotic cells, sensing DAMPs, and activating IFN signaling. They also express the highest level of pro-regenerative factors, *Cxcl10* and *Nampt*, suggesting a potential pro-regenerative role for these macrophages. This is supported by a previously published scRNA-seq study identifying a similar IFNRM subset in day 3 acutely injured muscle, which specifically expressed *Cxcl10*, an IFN-responsive gene, to promote MuSC proliferation [38]. That study also showed that blocking the binding of CXCL10 to its receptor in vivo resulted in poor muscle regeneration after acute muscle injury [38], supporting a pro-regenerative role for this molecule. Another published study showed that, in response to muscle injury, a subset of macrophages secrete NAMPT to stimulate the proliferation of myoblasts and therefore promote skeletal muscle regeneration in vivo [56]. By expressing a considerable level of *Igf1*, the MHCII^hi^ and TLF^+^ subclusters might also contribute to muscle regeneration, ECM remodeling, and fibrosis, as IGF-1 can exert both pro-regenerative [15,54,55] and pro-fibrotic functions [73,74]. Future studies with macrophage-specific targeting of the core signature genes of these macrophage subtypes are needed to verify their functions in injured muscles.

The activation and differentiation of monocytes/macrophage subpopulations appear mainly driven by the muscle tissue microenvironment. In uninjured muscle, the minor presence of IFNRMs may be induced by microinjuries in the steady state, and the MHCII^hi^ and TLF^+^ macrophages also appear to be important for maintaining muscle homeostasis based on their significant expression of *Igf1*. While barely detectable in uninjured muscle, monocytes are prominently present in injured muscle, indicating their active recruitment from blood circulation in response to injury. The monocyte fraction in acutely injured muscles, where injury is monophasic, peaks around day 3 and then recedes progressively, corresponding to inflammation resolution. A significant fraction of monocytes, however, remains in chronically injured dystrophic muscles where low-grade inflammation continues. Consistently, the Gpx3^+^ and Ccl8^+^ subclusters, which appear to promote inflammation resolution, are detected in acutely injured muscle after day 3, along with the injury repair, but not in *mdx^5cv^* muscles. The pro-inflammatory subcluster is primarily identified in day 0.5 to day 2 acutely injured muscles and it predominates the macrophage population in day 0.5 and day 1 muscles. It is barely detectable in later stage acutely injured muscles or in dystrophic muscles. This cluster is thus likely activated by massive muscle necrosis shortly after acute muscle injury. Interestingly, although the monocyte/macrophage subcluster compositions are similar between *mdx^5cv^* quadriceps and diaphragm, the proportions of some subclusters are different: the monocytes and TLF^+^ macrophage subcluster are bigger while the MHCII^hi^ subcluster is smaller in the diaphragm than in quadriceps. Unlike limb muscles including quadriceps, the muscle contraction in the diaphragm is constant, which likely induces higher mechanical stress, leading to more severe myofiber damage compared to quadriceps. This might trigger increased monocytes infiltration in the diaphragm than in quadriceps. We have previously reported that the embryonic non-hematopoietic stem cell-derived muscle resident macrophages, the TLF^+^ macrophages, were replaced over time by the adult bone marrow-derived resident macrophages, mainly the MHCII^hi^ macrophages [13]. This replacement is relatively delayed in the diaphragm than in quadriceps [13], which may contribute, in part, to the relatively larger proportion of the TLF^+^ subcluster in *mdx^5cv^* diaphragm than in *mdx^5cv^* quadriceps.

Both Ly6C^hi^ and Ly6C^lo^ intramuscular monocyte/macrophage subpopulations are heterogeneous, consisting of different subclusters in different injured muscles. But none of these subclusters conform to strict M1 or M2 activation based on their expression of the genes related to the M1 and M2 polarization, including those that encode cell surface makers, cytokines, arginine metabolism enzymes, and transcription factors. Therefore, the Ly6C^hi^/Ly6C^lo^ subtypes do not correlate with the M1/M2 activation status. Instead, the transcriptomes of macrophages change with the altered microenvironment in injured muscle, differentially expressing genes involved in inflammation, myofiber regeneration, and ECM remodeling. In line with our previous report that infiltrating macrophages are broadly activated, rather than bi-polarly activated, to support muscle injury repair [21], the current scRNAseq analysis adds additional evidence that the classical binary M1/M2 paradigm does not apply to the complex in vivo macrophage activation scenario in injured skeletal muscle.

The intramuscular monocyte/macrophage differentiation in injured muscles appears to be multi-directional, not linear. In a classic hypothetical scenario of linear differentiation, one would expect to see blood Ly6C^hi^ monocytes infiltrate skeletal muscle in response to injury and differentiate into Ly6C^hi^ pro-inflammatory macrophages. Ly6C^hi^ pro-inflammatory macrophages would then undergo phenotype switch to differentiate into Ly6C^lo^ macrophages that are anti-inflammatory, pro-fibrotic, and pro-regenerative. However, our analysis indicates that the pro-inflammatory subcluster predominates Ly6C^hi^ monocytes/macrophages only before day 3 post-acute injury, and that monocytes, but not the pro-inflammatory subcluster, comprise the majority of Ly6C^hi^ cells on day 3 post-acute injury when the IFNRMs start to emerge, along with the Gpnmb^+^Spp1^+^ and MHCII^hi^ macrophages. The pro-inflammatory subcluster is also barely detectable in 14-week-old *mdx^5cv^* muscles. Therefore, the IFNRM, Gpnmb^+^Spp1^+^, and MHCII^hi^ subclusters might not be differentiated from the pro-inflammatory subcluster, which is further supported by our pseudotime trajectory analysis. Together, these findings suggest a different monocytes differentiation route at day 3 post-acute injury and in 14-week-old *mdx^5cv^* muscles: blood Ly6C^hi^ monocytes directly differentiate into IFNRM, Gpnmb^+^Spp1^+^, and MHCII^hi^ macrophage subclusters, bypassing the pro-inflammatory stage. It remains unknown whether the pro-inflammatory subcluster on day 1 can differentiate into other Ly6C^lo^ clusters, and whether the Gpnmb^+^Spp1^+^ subcluster can further differentiate into MHCII^hi^ resident-like subcluster at later time points.

The multi-directional differentiation of intramuscular monocytes/macrophages is most likely driven by the diverse microenvironment within different injured muscle samples. There are abundant necrotic myofibers as well as a major influx of neutrophils on day 1 post-acute injury [14,15]. Necrotic cells are a rich source of DAMPs, which are potent stimuli driving the expression of pro-inflammatory genes through TLRs/NF-κB signaling by multiple cell types including monocytes/macrophages [98,99]. Meanwhile, infiltrating neutrophils produce a high level of pro-inflammatory mediators such as TNFα, IL-1β, and reactive oxygen species (ROS) [100], which can also potently stimulate the pro-inflammatory activation of monocytes/macrophages. On day 3 post-injury, the necrotic myofibers start to disappear via phagocytosis [14,15] and the fraction of neutrophils decreases significantly compared to day 1 post-injury. These changes are likely to cause a general loss of a potent pro-inflammatory microenvironment in injured muscle on day 3, in which the newly recruited monocytes undergo differentiation into macrophage subclusters other than the pro-inflammatory subcluster. In 14-week-old *mdx^5cv^* muscles, myofiber necrosis is patchy and not massive [19] and neutrophils are very few (Figure 1C), lacking a potent pro-inflammatory microenvironment. Consequently, the pro-inflammatory subcluster is barely identifiable in dystrophic muscles.

In summary, our study has identified multiple transcriptomically diverse monocyte/macrophage subclusters co-existing in each injured muscle sample. Both Ly6C^hi^ and Ly6C^lo^ macrophages are heterogenous, containing different subclusters which express the genes that are differentially involved in muscle inflammation, regeneration, and ECM remodeling. These subclusters do not conform to strict M1/M2 activation. The differentiation of intramuscular monocyte/macrophage clusters in injured muscles appears multi-directional and driven by specific microenvironment cues within injured muscles. The persistent monocyte/macrophage infiltration and subsequent differentiation induced by continuous injury may underlie the pathogenic roles of macrophages and contribute to dystrophic changes in *mdx^5cv^* muscles. Our study supports the notion that controlling muscle injury and subsequent macrophage infiltration and activation is important to the treatment of DMD.

## 4. Materials and Methods

### 4.1. Animals

*Wild-typed* (*WT*) *C57BL/6J* (Strain #:000664) and *mdx^5cv^* (Strain #:002379) mice were purchased from The Jackson Laboratory (Bar Harbor, ME, USA). Only males were used in this study because DMD only affect males. Our study protocol was approved by the Institutional Animal Care and Use Committee at Hospital for Special Surgery (New York, NY, USA).

### 4.2. Induction of Acute Skeletal Muscle Injury, and Collection of Muscle Samples

Fourteen-week-old male *WT* mice were used for the induction of acute skeletal muscle injury. BaCl_2_ (Millipore Sigma, Burlington, MA, USA) was dissolved in phosphate-buffered Saline (PBS, Fisher Scientific, Hampton, NH, USA) to a final concentration of 1.2% (*w*/*v*). A total of 100 µL of sterile BaCl_2_ solution was injected into the right quadriceps to induce acute skeletal muscle injury. One and three days post injection, the injured quadriceps was collected. Simultaneously, the quadriceps and diaphragm were collected from 14-week-old male *WT* and *mdx^5cv^* mice. Tendons were removed from the tissue specimens. Collected muscle samples were thoroughly washed in 1 × PBS (pH 7.4) to remove blood contamination.

### 4.3. Muscle Single-Cell Suspension Preparation

Muscle single-cell suspension was prepared by collagenase/dispase digestion. Each muscle was minced in 2.5 mL of digestion solution containing 1 U/mL of collagenase B and 1 U/mL of dispase II (Roche Diagnostics, Indianapolis, IN, USA) in PBS and incubated at 37 °C for 1 h. The reaction was terminated by addition of 10 mL of PBS with 10% of fetal bovine serum (FBS, Corning, Glendale, AZ, USA). The mixture was then filtered through 70-μm cell strainer and subjected to centrifugation at 250× *g* for 5 min. The pellet was collected, and the supernatant was centrifuged again at 250× *g* for 5 min at 4 °C. The pellets were combined, washed with PBS, and centrifuged at 670× *g* for 10 min. The pellet was resuspended in 3 mL of PBS, filtered through a 40-μm cell strainer, layered on an equal volume of Lymphocyte-M solution (Cedarlane, Burlington, NC, USA), and centrifuged at 2100× *g* for 45 min. Cells at the interface were collected in 10 mL PBS containing 10% FBS, centrifuged at 670× *g* for 10 min at 4 °C, and re-suspended in FACS buffer.

### 4.4. Flow Cytometry Analysis and Cell Sorting

Single-cell pellets were washed once in ice-cold FACS buffer (1 × PBS, 0.5% (*w*/*v*) bovine serum albumin (BSA, Millipore Sigma), 0.1% (*w*/*v*) sodium azide), centrifuged at 500× *g* for 3 min at 4 °C, and re-suspended in ice-cold FACS staining buffer (1 × PBS, 2% (*w*/*v*) BSA, 2% (*v*/*v*) normal mouse serum (Millipore Sigma).

For FACS analysis of intramuscular monocytes/macrophages, cells were stained on ice for 20 min with fluorescence-labeled antibodies targeting monocyte/macrophage specific markers. Monocytes/macrophages were identified as CD45^+^CD64^+^F4/80^+^ cells. Ly6C^hi^/Ly6C^lo^ monocytes/macrophages were further separated by simultaneous staining of Ly6C. Fluorescence-labeled corresponding normal IgG isotypes were included as negative controls for gating. All antibodies were from commercial providers and diluted for staining following manufacturer’s recommendations. The following antibodies were from BD Biosciences (Franklin Lakes, NJ, USA): PerCP-Cy5.5 Rat anti-mouse CD45 (Cat. #550994); PE Rat anti-mouse F4/80 (Cat. #565410); PE Rat IgG2a, κ Isotype Control (Cat. #559317); PE-Cy7 Rat IgG2a, κ Isotype Control (Cat. #552784); BV421 Rat IgG2a, κ Isotype Control (Cat. #562602); PerCP-Cy5.5 Rat IgG2b, κ Isotype Control (Cat. #550764). The following antibodies were from BioLegend (San Diego, CA, USA): BV421 Rat anti-mouse CD64 (Cat. #164407); PE-Cy7 Rat anti-mouse Ly6C (Cat. #128018). After staining, cells were washed twice with FACS buffer and analyzed on an LSR II (BD Biosciences, San Jose, CA, USA) with BD FACS Diva^TM^ software (v9.0, BD Biosciences). Collected data were then analyzed using FlowJo software (v10.10.0, Tree Star, Inc., Ashland, OR, USA).

To clean up single cell suspension for subsequent single-cell based analysis, cell sorting based on FSC/SSC was performed to exclude dead cells, cell duplexes, and tissue debris. Cell sorting was performed by the Flow Cytometry Core of the Boston University School of Medicine. Collected cells from 5 individual male mice were combined and pelleted by centrifuging at 300 × g for 10 min. For scRNA-seq analysis, pellets were re-suspended in FBS supplemented with 10% of DMSO and cryopreserved in liquid nitrogen.

### 4.5. Single-Cell cDNA Library Preparation and Sequencing

Cryopreserved cells were thawed in a 37 °C water bath for two minutes. The thawed content was added dropwise into 9 mL of warm freshly prepared RPMI 1640 (Gibco, Thermo Fisher Scientific, Waltham, MA, USA) with 20% of FBS, spun 300g for 5 min, and re-suspended in 5 mL of RPMI with 10% of FBS. The cells were counted, and viability was assessed using Acridine Orange and Propidium Iodide staining followed by counting on a K2 Cellometer (Nexcelome, Lawrence, MA, USA). Next, scRNA-seq was performed following the Single Cell 3′ Reagents Kits V3.1 User Guidelines (10× Genomics, Pleasanton, CA, USA). All samples were above 70% viability. Cells were loaded to target 5000 cells final for generating the single-cell emulsion (Chromium Single Cell 3′ Chip kit A v2 PN-12036 or v3 chip kit B PN-2000060). Reverse-transcription (RT) was performed in the emulsion, cDNA amplified, and library constructed with v3.1 chemistry. Libraries were indexed for multiplexing (Chromium i7 Multiplex kit PN-12062). Following preparation, libraries were quantified by Qubit 3 fluorometer (Invitrogen, Carlsbad, CA, USA) and quality was assessed by Bioanalyzer (Agilent Technologies, Santa Clara, CA, USA). Equivalent molar concentrations of libraries were pooled, and the reads were adjusted after sequencing the pools in a Miseq (Illumina, San Diego, CA, USA). The libraries were then sequenced in a Novaseq 6000 (Illumina) at the New York Genome Center (NYGC, New York, NY, USA) following 10× Genomics recommendations. 

### 4.6. Single Cell-Based mRNA Sequencing Analysis

External Datasets. Additional wild type macrophages were obtained from GSE142480. Acute injury macrophages were obtained from GSE138826.

Quality Control. The gene by cell expression matrices were loaded to the R package Seurat version 5.0 [101] for downstream analyses. Cells of low quality, defined by the detectable expression of fewer than 200 genes per 1000 Unique Molecular Identifiers (UMI) and greater than 7.5% mitochondrial content, were filtered out. To verify the composition of the filtered dataset, automated annotation of main cell types was performed using SingleR with the ImmGen reference set [30,31].

Normalization. Each cell was scored for cell cycle using the mouse orthologs of cell cycle genes for S and G2M phase using the Seurat command “CellCycleScoring” [102]. The human cell cycle genes were converted to mouse orthologs using the R package gprofiler2 command “gorth” [103]. The difference between S and G2M phase scores was calculated for use during normalization. The data was normalized and scaled while regressing out cell cycle difference and mitochondrial content using the glmGamPoi implementation of the Seurat “SCTransform” function [104]. PCA was performed on the expression. The SNN graph and clusters were found with the Seurat functions “FindClusters” and “FindNeighbors” using the first 10 PCs.

Monocyte/Macrophage Filtering. Clusters showing high expression of the monocyte/macrophage markers Adgre1 and Fcgr1 were selected, and the Seurat “merge” function was used to pool those clusters together. Cells with normalized expression of the monocyte/macrophage markers Adgre1 and Fcgr1 less than one and fibro adipogenic progenitor (FAP) cell markers Pdgfra and Ly6a greater than zero were filtered out. The new pooled monocytes/macrophages were then renormalized using the normalization specified above.

Batch Correction. Given the technical variation in both GSE142480 and GSE138826, batch correction and integration were performed using Harmony [105]. Since Harmony is included as an option for batch correction in the Seurat package, Harmony was run using default parameters on the PCA reduction of the data. For all downstream analyses only the integrated dataset combining GSE142480 and the collected dataset was used.

Pseudotime Trajectory. Monocle 3 and Slingshot were used to perform pseudotime analysis on the pooled MO/MPs [106,107]. For both methods the starting cluster was determined by the highest expression of MO/MP maturity marker gene Ly6c2.

Monocle 3. The Seurat object was converted to a CellDataSet using the Monocle function “as.cell_data_set”. The CellDataSetObject was then clustered and the graph was learned using the Monocle functions “cluster_cells” and “learn_graph”. The cells were then ordered in pseudotime using the Monocle function “order_cells”.

Slingshot. The Seurat was converted to a SingleCellExperiment object using the seurat function “as.SingleCellExperiment”. The “slingshot” function was run on the Harmony dimensionality reduction with the cells divided into two clusters: starting cluster and all other cells.

Differential Gene Expression. Positively and negatively differentially expressed genes were found for each cluster relative to all other clusters using the Seurat function FindAllMarkers. The function parameters min.pct were set to 0.25 and min.diff.pct were set to 0.1.

Functional Enrichment. Gene Set Enrichment Analysis was performed using WebGestalt [108]. A rank file was generated by calculating the log2 fold change (log2FC) for all genes using the Seurat function FoldChange. The genes were then sorted in descending order by log2FC. Webgestalt was run using the Gene Ontology biological processes (GOBP) gene sets at an FDR of 0.05 [109].

### 4.7. Statistical Analysis

Data were analyzed using GraphPad Prism 10 software (v10, GraphPad Software, San Diego, CA, USA). The Mann–Whitney test was performed to compare between the two groups; the Kruskal–Wallis test followed by Dunn’s test was performed to compare multiple (≥3) groups. A *p* value of <0.05 was considered statistically significant.

## Figures and Tables

**Figure 1 ijms-26-08098-f001:**
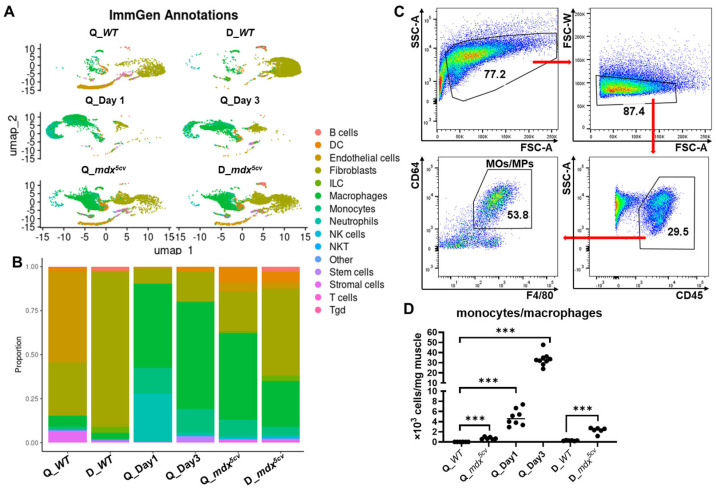
Single-cell-based RNA sequencing (scRNAseq) analysis reveals that monocytes/macrophages expand dramatically in both acutely and chronically injured skeletal muscle compared to uninjured muscle. The skeletal muscle single-cell suspension was prepared from the quadriceps (Q_*WT*) and diaphragm (D_*WT*) of 14-week-old uninjured male *WT* mice, the quadriceps of 14-week-old male *WT* mice at day 1 (Q_Day1) and day 3 (Q_Day3) post-acute injury, and the quadriceps (Q_*mdx^5cv^*) and diaphragm (D_*mdx^5cv^*) of 14-week-old male *mdx^5cv^* mice. scRNAseq analysis was performed with single-cell suspension prepared and combined from 5 mice/group. (**A**) The automated annotation of main cell types was performed using SingleR with the ImmGen reference set. DC: dendritic cell. ILC: innate lymphoid cell. NK cells: nature killer cells. NKT: nature killer/T cell. Tgd: gamma-delta T cells. (**B**) Bar graph showing the proportion of different cell types. (**C**) Representative dot plots showing the gating strategy of intramuscular monocytes/macrophages. The data shown was from the diaphragm of 14-week-old *mdx^5cv^* mice. (**D**) Intramuscular density of total monocytes/macrophages determined by flow cytometry (FACS) analysis. N ≥ 6 mice/group. *** *p* ≤ 0.001.

**Figure 2 ijms-26-08098-f002:**
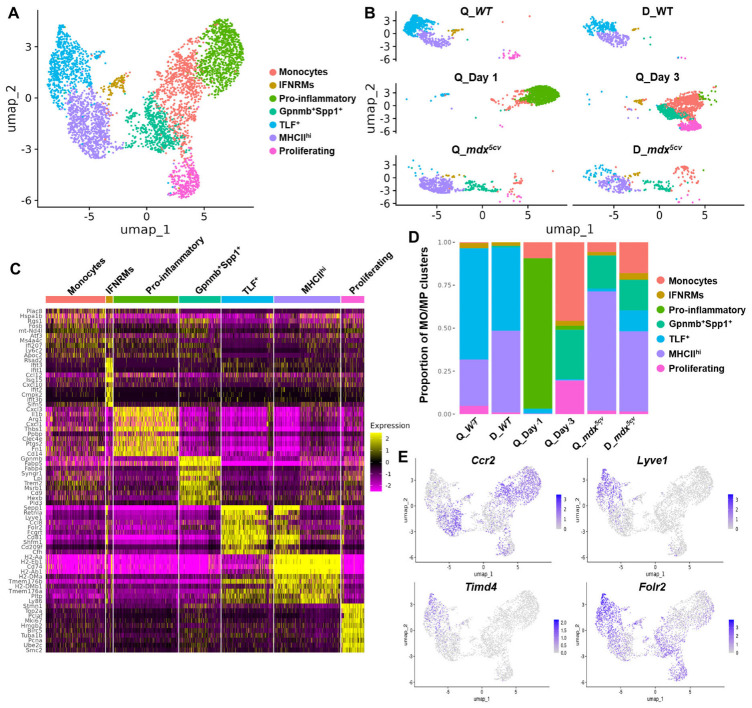
Monocytes/macrophages in uninjured, acutely injured, and chronically injured skeletal muscle contain multiple subclusters with diverse transcriptomes. Monocyte/macrophage scRNAseq data from Q_Day 1, Q_Day 3, Q_*mdx^5cv^*, and D_*mdx^5cv^* were extracted, pooled and integrated with previously published macrophage scRNAseq data from *WT* quadriceps and diaphragm (GEO: GSE142480). (**A**) UMAP of pooled and integrated monocyte/macrophage scRNAseq data identifying subclusters named as indicated. (**B**) UMAP showing monocyte/macrophage subclusters in each individual sample. (**C**) Heatmap of top 10 differentially expressed genes (DEGs) of each monocyte/macrophage subcluster identified. (**D**) Proportion of monocyte/macrophage (MO/MP) subcluster to total monocytes/macrophages in each muscle sample calculated based on the number of each subcluster identified in individual muscle samples. (**E**) Feature plots showing the expression of Ccr2, Lyve1, Timd4, and Folr2 by pooled and integrated monocytes/macrophages.

**Figure 3 ijms-26-08098-f003:**
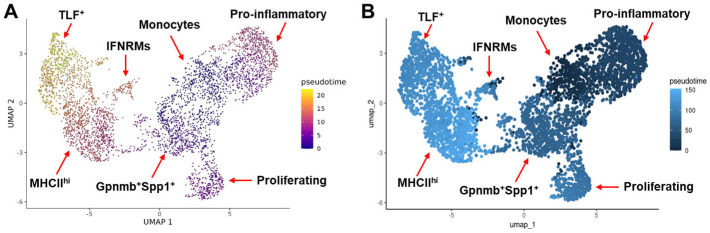
Pseudotime analysis infers the differentiation trajectory of monocyte/macrophage subclusters. Analysis was performed with both Monocle 3 (**A**) and Slingshot (**B**).

**Figure 4 ijms-26-08098-f004:**
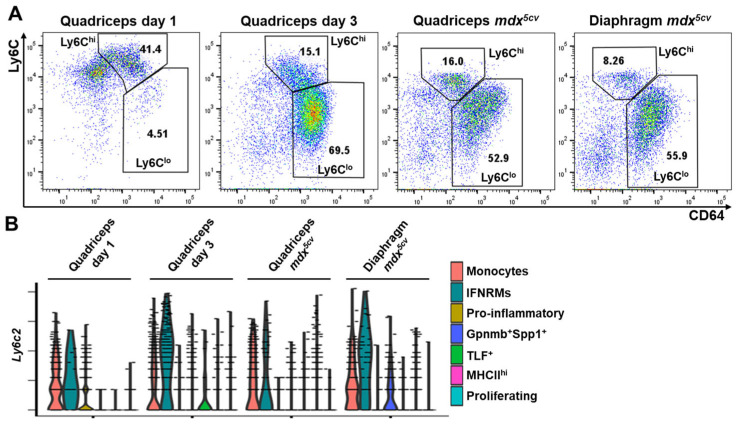
Both Ly6C^hi^ and Ly6C^lo^ intramuscular macrophages are heterogenous. (**A**) Flow cytometry (FACS) analysis identifying Ly6C^hi^ and Ly6C^lo^ monocytes/macrophages (CD64^+^) in acutely and chronically injured skeletal muscles. Dot plots showing the expression of CD64 and Ly6C by gated CD45^+^ cells. N ≥ 5 mice/group. (**B**) Violin plots of scRNAseq analysis showing the expression of the *Ly6c2* gene by different monocyte/macrophage subclusters in each injured muscle sample.

**Figure 5 ijms-26-08098-f005:**
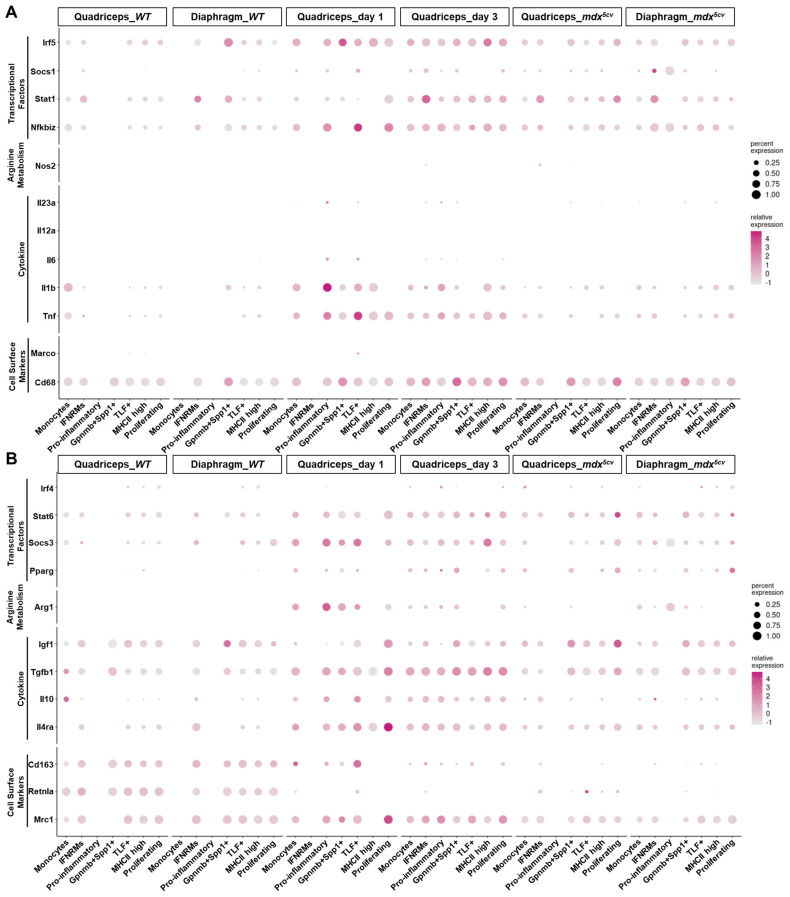
Monocyte/macrophage subclusters in injured skeletal muscle do not conform to strict M1 or M2 activation. Dot plots showing the expression of M1 (**A**) and M2 (**B**) genes by different monocyte/macrophage clusters.

**Figure 6 ijms-26-08098-f006:**
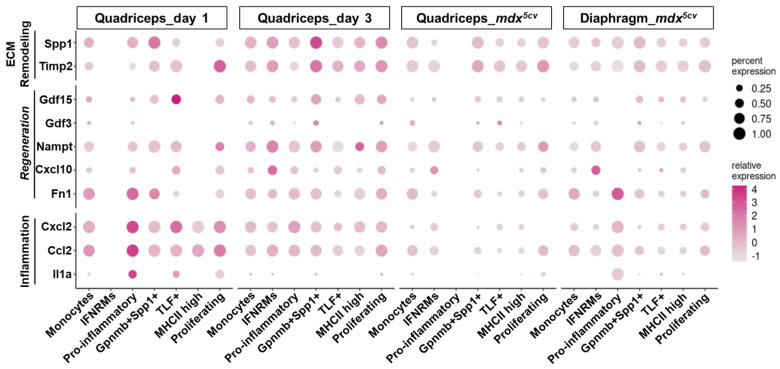
Monocyte/macrophage subclusters express genes differentially involved in inflammation, regeneration, and ECM remodeling, in injured skeletal muscle. Dot plots showing the expression of genes involved in inflammation, regeneration, and ECM remodeling by different monocyte/macrophage clusters.

**Figure 7 ijms-26-08098-f007:**
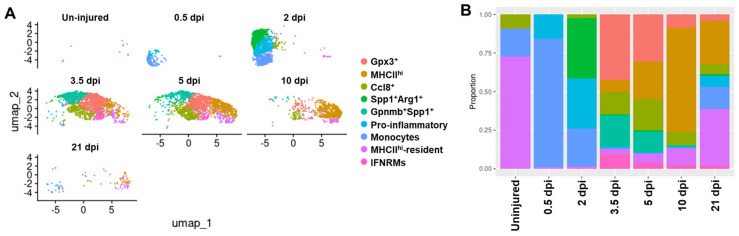
Gpnmb^+^Spp1^+^ macrophage subcluster disappears along with emergence of potentially pro-resolution macrophage subclusters at late stages of acute skeletal muscle injury repair. (**A**) UMAP analysis showing monocyte/macrophage clusters in each muscle sample as labeled. (**B**) Proportion of individual monocyte/macrophage (MO/MP) clusters to total monocytes/macrophages in each muscle sample. dpi: days post injury.

## Data Availability

New single-cell based RNA sequencing (scRNA-seq) data were generated in this study. Data for Quadriceps_*WT*, Diaphragm_*WT*, Quadriceps_*mdx^5cv^*, and Diaphragm_*mdx^5cv^* had been deposited and publicly available at GEO (GSE218201) along with a previous publication [64]. Quadriceps_Day 1 and Quadriceps_Day 3 have been separately deposited at GEO (GSE270031) and are publicly available as of the date of publication.

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
