# Peer review of "Heterogeneous Macrophage Activation in Acute Skeletal Muscle Sterile Injury and mdx5cv Model of Muscular Dystrophy"

_ijms, 2025, doi:10.3390/ijms26168098_

Round 1
Reviewer 1 Report
Comments and Suggestions for Authors
The manuscript by Wang et al. provides a valuable single-cell resource profiling of macrophage heterogeneity in uninjured, acutely injured, and chronically injured (mdx5cv) mouse muscle. The dataset is carefully generated, and the main conclusion that a persistent, Spp1-high macrophage state correlates with chronic fibrosis in dystrophic muscle is plausible and agrees with prior work that identified similar Spp1⁺/Trem2⁺ “scar-associated” macrophages in dystrophic (PMID: 37418531), and ageing muscle (PMID: 39103421), as well as other fibrotic tissues across injury models. However, the manuscript does not yet integrate fully with the rapidly expanding literature nor provide any functional evidence that the identified subset is causal rather than correlative. If the authors are not able to functionally test the inferred pathogenicity of this subset, they should at least do a better job comparing the populations with other publicly available scRNA-seq datasets in the literature and use more consistent nomenclature.
Major points
- Previous loss-of-function studies show that global or macrophage-specific deletion of Spp1 ameliorates mdx pathology and intramuscular fat accumulation (PMID: 40626359). Other studies also demonstrate that TREM2 blockade or deficiency limits fibrosis in the lung (PMID: 39971937) and modulates macrophage-dependent cardiac repair after myocardial infarction (PMID: 38182899). The authors briefly summarize some of these studies in the discussion, but should discuss the data earlier, clarify remaining mechanistic gaps, and temper the “causality” language, especially in the absence of any perturbations (i.e., the use of KO).
- The Discussion omits several other key studies and datasets. Incorporating these studies will demonstrate the conservation of macrophage states across species, injuries, and disease contexts. For example:
- The Nagy lab recently revealed multilayered regenerative-inflammation zones (RIZs) and GPNMB⁺ macrophage subsets in similar acute and chronic injury models (PMID 39190487). They also did a subset comparison between DMD and acute models.
- The Miceli lab (PMID 36123393) used single-nuclei transcriptomics to demonstrate that dystrophin rescue normalizes an “MDSC-like” myeloid population and restores repair/remodelling macrophages. Is this subpopulation and interpretation based on markers relevant in the mdx5cv model?
- Human skeletal-muscle single-cell atlases that report analogous SPP1⁺/TREM2⁺ macrophages in adult tissue and trauma (PMID: 32624006 and others).
- Several clusters are labelled “pro-regenerative” or “anti-inflammatory” solely by one or two markers. A more expanded comparison of the authors’ marker list (Cd9, Gpnmb, Fabp5, Lgals3, Trem2) with published “scar-associated”, “fibrogenic”, “growth factor-expressing (GFEMs)”, “MDSC-like” signatures is needed, ideally supplied as a correlation heatmap or supplementary table (i.e., top 20 genes per cluster versus published signatures to substantiate cluster names).
- In line with my previous comment, the manuscript retains the binary M1/M2 paradigm, which single-cell studies have largely abandoned. The authors rightfully conclude that the classical paradigm doesn’t apply here, but I would suggest they drop any “M1/M2” labels and references to this paradigm and use contemporary terms such as “pro-inflammatory”, “IFN-responsive”, “ECM-remodeling”, “growth factor expressing”, or “resolution-phase” for full consistency with single-cell nomenclature.
- Explain the choice of the term “Spp1⁺” over alternatives such as “Trem2⁺” or “Gpnmb+” (top Marker in their dataset) and try to align nomenclature with the current field consensus.
- Tissue immunofluorescence validation and spatial localization of predicted cell states would be valuable. Otherwise, any designations that are not experimentally validated, nor are they supported by protein expression, cytokine measurements, or in situ localization, add more confusion to the field... My recommendation to the authors is to base their naming on other published studies for consistency and clarity.
- Provide fuller details of batch-correction, external-dataset integration, and quality-control thresholds, and which pseudotime workflow was used, so that readers can reproduce the analysis. Ideally, use a second pseudotime pipeline to validate the findings in Fig. 3C. Slingshot, dynamic modelling from scVelo, Monocle, etc, are generally good trajectory inference options.
Minor points
- A more specific title that mentions the mdx5cv model and acute injury (e.g., “…in acute sterile injury and mdx5cv model of dystrophy”) would help non-specialist readers.
- Several figures are misaligned and cut outside of margins (e.g. Fig. 2B).
- FACS plots should indicate the start of the axes.
- Figures 5 and 6 are not very reader-friendly. I would recommend using a different visualization (i.e., dot plot).
- Major cell type predictions should be cross-referenced with databases like Immgen. Please try adding an automated annotation pipeline, such as SingleR, which will provide a cell type prediction confidence score.
- Conclusions about the embryonic versus monocyte origin based solely on Timd4, Lyve1, or Ccr2 expression in the context of muscle are suggestive but limited, and not definitive. Please qualify any lineage statements unless supported by fate-mapping data.
Author Response
Major points:
Major Point #1: Previous loss-of-function studies show that global or macrophage-specific deletion of Spp1 ameliorates mdx pathology and intramuscular fat accumulation (PMID: 40626359). Other studies also demonstrate that TREM2 blockade or deficiency limits fibrosis in the lung (PMID: 39971937) and modulates macrophage-dependent cardiac repair after myocardial infarction (PMID: 38182899). The authors briefly summarize some of these studies in the discussion, but should discuss the data earlier, clarify remaining mechanistic gaps, and temper the “causality” language, especially in the absence of any perturbations (i.e., the use of KO).
Author Response : Thank you for the comments and suggestions. We have moved up the discussion of Gpnmb+Spp1+ macrophage subclusters, added the discussion of the findings from the papers mentioned, and cited these papers (Line 504-556). We have also tempered the causality statement.
Major Point #2: The Discussion omits several other key studies and datasets. Incorporating these studies will demonstrate the conservation of macrophage states across species, injuries, and disease contexts. For example:
-
- The Nagy lab recently revealed multilayered regenerative-inflammation zones (RIZs) and GPNMB⁺ macrophage subsets in similar acute and chronic injury models (PMID 39190487). They also did a subset comparison between DMD and acute models.
Author Response: Thank you for pointing this out. We have cited and discussed this work in both Results and Discussion sections (Line 179-187, 546-554).
- The Miceli lab (PMID 36123393) used single-nuclei transcriptomics to demonstrate that dystrophin rescue normalizes an “MDSC-like” myeloid population and restores repair/remodeling macrophages. Is this subpopulation and interpretation based on markers relevant in the mdx5cv model?
Author Response: Our single-cell transcriptomics did not identify macrophage subcluster resembling “MDSC-like” population as reported by the mentioned reference. We have cited this work in Discussion. (Line 554-556).
- Human skeletal-muscle single-cell atlases that report analogous SPP1⁺/TREM2⁺ macrophages in adult tissue and trauma (PMID: 32624006 and others).
Author Response: The reference referred does not report SPP1⁺/TREM2⁺ macrophages. We have cited other work (Line 554-556)
Major Point #3: Several clusters are labelled “pro-regenerative” or “anti-inflammatory” solely by one or two markers. A more expanded comparison of the authors’ marker list (Cd9, Gpnmb, Fabp5, Lgals3, Trem2) with published “scar-associated”, “fibrogenic”, “growth factor-expressing (GFEMs)”, “MDSC-like” signatures is needed, ideally supplied as a correlation heatmap or supplementary table (i.e., top 20 genes per cluster versus published signatures to substantiate cluster names).
Author Response: We have revised this part with a more expanded comparison in Results section (2.2) accordingly. (Line 152-189, 220-227)
Major Point #4: In line with my previous comment, the manuscript retains the binary M1/M2 paradigm, which single-cell studies have largely abandoned. The authors rightfully conclude that the classical paradigm doesn’t apply here, but I would suggest they drop any “M1/M2” labels and references to this paradigm and use contemporary terms such as “pro-inflammatory”, “IFN-responsive”, “ECM-remodeling”, “growth factor expressing”, or “resolution-phase” for full consistency with single-cell nomenclature.
Author Response: We completely agree with this comment. However, this over-simplified paradigm is still being applied by some of the recent publications, including some referred here by the Reviewer. We include the analysis of M1/M2 gene expression by monocyte/macrophage subclusters to indicate and emphasize the inappropriateness of applying this binary macrophage activation paradigm to the complex in vivo scenario. We have revised related results and discussion part to better serve our purpose (Line 328-335, 615-622).
Major Point #5: Explain the choice of the term “Spp1⁺” over alternatives such as “Trem2⁺” or “Gpnmb+” (top Marker in their dataset) and try to align nomenclature with the current field consensus.
Major Point #6: Tissue immunofluorescence validation and spatial localization of predicted cell states would be valuable. Otherwise, any designations that are not experimentally validated, nor are they supported by protein expression, cytokine measurements, or in situ localization, add more confusion to the field... My recommendation to the authors is to base their naming on other published studies for consistency and clarity.
Author Response to Point #5 and #6: We have renamed “Spp1+” as “Gpnmb+Spp1+” to reflect that Gpnmb is the top marker of this macrophage subcluster, and to align with published study (ref. 36, 37). We keep “Spp1+” as this is a widely used term for reports of scar-associated fibrogenic macrophages (ref. 32-34), which are highly similar to the “Gpnmb+Spp1+” cluster identified by our study, and is among the top 3 markers of this cluster. We have also renamed “IFN-activated” as “IFN-responsive macrophage (IFNRM)” which has been used by a published study (ref. 38)
Major Point #7: Provide fuller details of batch-correction, external-dataset integration, and quality-control thresholds, and which pseudotime workflow was used, so that readers can reproduce the analysis. Ideally, use a second pseudotime pipeline to validate the findings in Fig. 3C. Slingshot, dynamic modelling from scVelo, Monocle, etc, are generally good trajectory inference options.
Author Response: We have revised the method part of scRNAseq accordingly (Section 4.6, Line 746-794). We have included the analysis with both Monocle3 and Slingshot tools (New Figure 3A and 3B).
Minor points:
Minor Point #1: A more specific title that mentions the mdx5cv model and acute injury (e.g., “…in acute sterile injury and mdx5cv model of dystrophy”) would help non-specialist readers.
Author Response: We have changed the title to “Heterogeneous macrophage activation in acute skeletal muscle sterile injury and mdx5cv model of muscular dystrophy”.
Minor Point #2: Several figures are misaligned and cut outside of margins (e.g. Fig. 2B).
Author Response: Corrected.
Minor Point #3: FACS plots should indicate the start of the axes.
Author Response: We have revised accordingly (Figure 1C and 4A).
Minor Point #4: Figures 5 and 6 are not very reader-friendly. I would recommend using a different visualization (i.e., dot plot).
Author Response: We have replaced both Figure 5 and 6 with dot plots.
Minor Point #5: Major cell type predictions should be cross-referenced with databases like Immgen. Please try adding an automated annotation pipeline, such as SingleR, which will provide a cell type prediction confidence score.
Author Response: We have replaced original Figure 1A-1C with new Figure 1A and 1B, which are generated by automated annotation using SingleR with the ImmGen reference set.
Minor Point #6: Conclusions about the embryonic versus monocyte origin based solely on Timd4, Lyve1, or Ccr2 expression in the context of muscle are suggestive but limited, and not definitive. Please qualify any lineage statements unless supported by fate-mapping data.
Author Response: We have revised it accordingly (Line 252-255).
Reviewer 2 Report
Comments and Suggestions for Authors
This manuscript presents a comprehensive analysis of macrophage subtypes in the limb muscle at various physiological states: normal non-injured state, acutely injured state, and chronically injured state. They also compared normal diaphragm and chronically injured diaphragm muscles. The results provide novel insights into the context-dependent distinct roles of macrophages in muscle repair and fibrosis. The study is based on single cell RNA-seq analysis with sample groups that have never been done previously. The methods are state-of-art and the data are interpreted properly. I have only a few minor comments:
- The scRNA-seq initially identified various cell clusters, and the monocyte/macrophage cluster is further examined and clustered in various subsets. Both original cluster and the subset are called clusters, which may be confusion to readers. I suggest that a clear nomenclature be used to avoid the potential confusion. The use of subset or subcluster should effectively avoid such confusion.
- The both limb and diaphragm muscle are used. Due to their distinct usage patterns and physiological functions, it would be interesting to tie of the results into the physiological function of these muscles in the discussion section.
-
Figure 6, "Monocyte/macrophage clusters contribute differently to inflammation, extracellular matrix (ECM) remodeling, and myogenesis in injured skeletal muscle". This is only based on gene signature without any functional validation. I suggest that this sentence be modified to avoid overstatement of the results. This issue also appears in other occasions where the gene signature is interpreted as cell function. Such over-interpretations should be carefully avoided and the lack of functional validation should be discussed as a weakness in the discussion.
- In Figure 7, what is the cluster (subset) of "pro-inflammation resolution macrophages"? are they both pro-inflammatory and anti-inflammatory?
Author Response
Major Comments: None
Minor Comments:
Minor Comment #1: The scRNA-seq initially identified various cell clusters, and the monocyte/macrophage cluster is further examined and clustered in various subsets. Both original cluster and the subset are called clusters, which may be confusion to readers. I suggest that a clear nomenclature be used to avoid potential confusion. The use of subset or subcluster should effectively avoid such confusion.
Author Response: We have changed to “subcluster” for subset of macrophages.
Minor Comment #2: Both limb and diaphragm muscle are used. Due to their distinct usage patterns and physiological functions, it would be interesting to tie the results into the physiological function of these muscles in the discussion section.
Author Response: We have added discussion of this (Line 597-609).
Minor Comment #3: Figure 6, "Monocyte/macrophage clusters contribute differently to inflammation, extracellular matrix (ECM) remodeling, and myogenesis in injured skeletal muscle". This is only based on gene signature without any functional validation. I suggest that this sentence be modified to avoid overstatement of the results. This issue also appears in other occasions where the gene signature is interpreted as cell function. Such over-interpretations should be carefully avoided and the lack of functional validation should be discussed as a weakness in the discussion.
Author Response: We agree and have modified as “Monocyte/macrophage subclusters express genes differentially involved in inflammation, re-generation, and ECM remodeling, in injured skeletal muscle” (Line 383-384). We also replaced cell function claim with description of gene signature in other occasions.
Minor Comment #4: In Figure 7, what is the cluster (subset) of "pro-inflammation resolution macrophages"? Are they both pro-inflammatory and anti-inflammatory?
Author Response: Thank you for pointing this out. We have changed the term to “pro-resolution macrophages”. (line 483-484)
Round 2
Reviewer 1 Report
Comments and Suggestions for Authors
The authors did a good job addressing my comments. One minor observation is to ask the authors to please double-check the cluster ID names and tabs in the Supplementary tables (S2) for accuracy with the new nomenclature used in the revised version.